# Combined Reporting of Surgical Quality and Cancer Control after Surgical Treatment for Penile Tumors with Inguinal Lymph Node Dissection: The *Tetrafecta* Achievement

Aldo Brassetti [1,*], Umberto Anceschi [1], Gabriele Cozzi [2], Julian Chavarriaga [3,4], Pavel Gavrilov [5], Josep Maria Gaya Sopena [5], Alfredo Maria Bove [1], Francesco Prata [1], Mariaconsiglia Ferriero [1], Riccardo Mastroianni [1], Leonardo Misuraca [1], Gabriele Tuderti [1], Giulia Torregiani [6], Marco Covotta [6], Diego Camacho [7], Gennaro Musi [2], Rodolfo Varela [7], Alberto Breda [5], Ottavio De Cobelli [2] and Giuseppe Simone [1]

1   Department of Urology, IRCCS "Regina Elena" National Cancer Institute, 00144 Rome, Italy
2   Department of Urology, European Institute of Oncology, 20141 Milan, Italy
3   Division of Urology, Clinica Imbanaco, Quiron Salud, Cali 760042, Colombia
4   Division of Urology, Pontificia Universidad Javeriana, Bogota 110231, Colombia
5   Department of Urology, Fondacio Puigvert, 08025 Barcelona, Spain
6   Department of Anesthesiology, IRCCS "Regina Elena" National Cancer Institute, 00144 Rome, Italy
7   Division of Urologic Oncology Instituto Nacional de Cancerologia, Bogota 111511, Colombia
*   Correspondence: aldo.brassetti@gmail.com; Tel.: +39-0652666772

**Abstract:** Background: To optimize results reporting after penile cancer (PC) surgery, we proposed a *Tetrafecta* and assessed its ability to predict overall survival (OS) probabilities. Methods: A purpose-built multicenter, multi-national database was queried for stage I–IIIB PC, requiring inguinal lymphadenectomy (ILND), from 2015 onwards. Kaplan–Meier (KM) method assessed differences in OS between patients achieving *Tetrafecta* or not. Univariable and multivariable regression analyses identified its predictors. Results: A total of 154 patients were included in the analysis. The 45 patients (29%) that achieved the *Tetrafecta* were younger (59 vs. 62 years; $p = 0.01$) and presented with fewer comorbidities (ASA score $\geq$ 3: 0% vs. 24%; $p < 0.001$). Although indicated, ILND was omitted in 8 cases (5%), while in 16, a modified template was properly used. Although median LNs yield was 17 (IQR: 11–27), 35% of the patients had <7 nodes retrieved from the groin. At Kaplan–Maier analysis, the *Tetrafecta* cohort displayed significantly higher OS probabilities (Log Rank = 0.01). Uni- and multivariable logistic regression analyses identified age as the only independent predictor of *Tetrafecta* achievement (OR: 0.97; 95%CI: 0.94–0.99; $p = 0.04$). Conclusions: Our *Tetrafecta* is the first combined outcome to comprehensively report results after PC surgery. It is widely applicable, based on standardized and reproducible variables and it predicts all-cause mortality.

**Keywords:** penile cancer; inguinal lymphadenectomy; Tetrafecta; surgical quality; survival

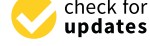



## 1. Introduction

Penile cancer (PC) is rare, especially in Europe and United States [1], while it represents up to 6% of all malignancies in men from developing countries of South America, Africa, and Asia. This uneven incidence is ascribed to a limited rate of neonatal circumcision in such parts of the world, which is considered as a protective factor against the disease [2]. Most of PC (95%) are squamous cell carcinomas (SCC) and surgery represents the mainstay of treatment for this disease, being primary tumor excision indicated both in localized and locally advanced cases [3]. It is an aggressive and mutilating neoplasm, whose most common sites of metastasis are locoregional lymph nodes (LNs) [4]. Tumor spread occurs in a consistent pattern, first involving inguinal (ILNs) and subsequently pelvic LNs (PLNs), in accordance with the route of anatomical drainage [5].

As approximately a quarter of patients with impalpable ILNs have micro-metastases, surveillance is only recommended in pTa/pTis tumors and pT1G1, owing to them having a comparatively low risk of lymphatic spread. Invasive nodal staging, which can be achieved by either dynamic sentinel-node biopsy or by modified ILNs dissection (ILND) (mILND), is mandatory for pT1 intermediate/high risk (pT1IR/HR) diseases as well as pT2–T4 tumors, whereas all patients with palpable inguinal nodes (cN1-3) should be offered radical ILND (rILND) [3]. Despite clear recommendations supporting these potentially life-saving treatments, several studies highlighted low guideline adherence [6–8] which may be attributable to the anticipated morbidity due to impaired lymph drainage from legs and scrotum, that can be as high as 50% [9].

To help standardizing outcomes reporting after surgery, the concept of "trifecta" was first proposed in the field of prostate-cancer treatment [10] and further extended to many other uro-oncological procedures [11,12]. In the present paper, we propose a novel method, based on combined reproducible variables, to report global results after PC surgery with ILND.

## 2. Materials and Methods

### 2.1. Study Design

After institutional board approval, we retrospectively reviewed data of patients treated for AJCC stage I-IIIB PC [13] at the four participating centers, from 2015 onwards. Only SCC of the penis were included in the study. Men presenting with a low risk of nodal tumor involvement, in which ILND was not indicated [3], were excluded from the analysis. The following parameters were collected:

- Patients' baseline characteristics (age, body mass index [BMI], American Society of Anesthesiologists [ASA] score);
- Clinical tumor stage [14];
- Surgical technique used for the treatment of both primary tumor and ILNs
- Final histology, number of retrieved LNs, local surgical margin status, pathologic tumor stage (defined according to the TNM classification and the American Joint Committee on Cancer stratification systems [14];
- Post-operative complications (stratified according to the Clavien-Dindo [CD] scale) [15].
- Recurrence-free (RFS), cancer-specific (CSS), and overall survivals (OS).

### 2.2. Tetrafecta Definition and Study Objective

The *Tetrafecta* for PC-surgery was conceived combining four standardized and reproducible variables: negative local surgical margins (NSM), no complications CD grade $\geq 3$, $\geq 7$ LNs retrieved from each treated groin [16], no evidence of disease at 12 months (NED12mo). We assessed the ability of this novel composite outcome to predict OS probabilities.

### 2.3. Statistical Analysis

The study population was split according to *Tetrafecta* achievement. Frequencies and proportions were used to report categorical variables, which were compared by means of the $\chi^2$-test. Continuous variables were presented as median and interquartile ranges (IQRs) and were compared using either the Mann Whitney U test or the Kruskal Wallis one-way, based on their normal or not-normal distribution, respectively (normality of the distribution of variables was tested by the Kolmogorov Smirnov test). The Kaplan–Meier method was used to assess the role of *Tetrafecta* achievement in predicting OS probabilities. Survival rates were computed at 2 and 5 years after surgery, and the log-rank test was applied to assess statistical significance between the two groups. Univariable and multivariable logistic regression analyses were used to identify predictors of *Tetrafecta* achievement. Significance level was set at a *p* value of <0.05. Statistical analysis was performed using the Statistical Package for Social Science v. 24.0 (IBM, Somers, NY, USA).

### 3. Results

Overall, data concerning 185 patients were collected; among these, 21 were excluded because of a <12 months follow-up and 10 for a diagnosis of non-SCC at final pathology. Out of the 154 included men, 45 (29%) achieved the *Tetrafecta* (Figure 1) and these were significantly younger (59 vs. 62 years; *p* = 0.01) and presented with fewer comorbidities (ASA score ≥ 3: 0% vs. 24%; *p* < 0.001) (Table 1).

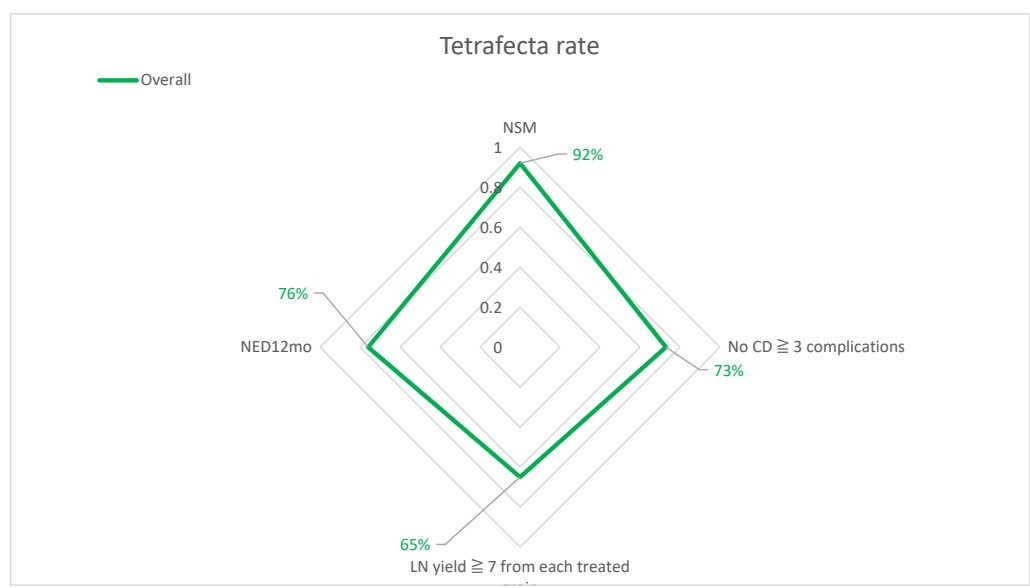

**Figure 1.** Surgical outcomes and *Tetrafecta* rate.

**Table 1.** Patients' characteristics and outcomes after PC surgery, according *Tetrafecta* achievement.

| | Overall <br> *n* = 154 | No *Tetrafecta* <br> *n* = 109 (71%) | *Tetrafecta* <br> *n* = 45 (29%) | *p* |
|---|---|---|---|---|
| **Age, years** | **61 (52–69)** | **62 (54–71)** | **59 (49–64)** | **0.01** |
| **BMI** | **27 (24.9–30.5)** | **27.1 (24.9–30.9)** | **26.9 (24.1–28.8)** | **0.05** |
| **Obesity, *n* (%)** | **45 (29%)** | **36 (33%)** | **9 (20%)** | **0.11** |
| **ASA score ≥ 3, *n* (%)** | **24 (16%)** | **24 (22%)** | **0 (0%)** | **<0.001** |
| **Clinical Tumor Stage, *n* (%)** | | | | |
| - T1 G ≥ 2 | 6 (10%) | 9 (8%) | 7 (16%) | 0.19 |
| - T2–3 and/or cN+ | 138 (90%) | 100 (92%) | 38 (84%) | |
| **Partial penectomy, *n* (%)** | 122 (79%) | 86 (79%) | 36 (80%) | 0.88 |
| - Penile reconstruction *, *n* (%) | 63 (51%) | 42 (38%) | 21 (47%) | 0.62 |
| **ILND, *n* (%)** | | | | |
| - None | 8 (5%) | 8 (7%) | 0 (0%) | |
| - Modified | 16 (10%) | 9 (8%) | 7 (16%) | 0.46 |
| - Radical | 130 (85%) | 92 (85%) | 38 (84%) | |
| **LN yield ‡, *n*** | 17 (11–27) | 14 (8–25) | 25 (17–32) | <0.001 |
| **LOS, d** | 3 (3–5) | 3 (3–5) | 3 (3–5) | 0.69 |
| **pAJCC Stage Groups †, *n*** | | | | |
| - 0–II | 51 (35%) | 35 (35%) | 16 (35%) | |
| - IIIa–IIIb | 41 (28%) | 29 (29%) | 12 (27%) | 0.47 |
| - IV | 54 (37%) | 37 (36%) | 17 (38%) | |

Data are reported as Median(IQR); * data refer to 122 patients (79%) that received partial penectomy; ‡ data refer to 146 patients (95%) that received ILND; † data refer to 146 patients (95%) as the proper pAJCC stage could not be assigned to the 8 (5%) pNx; BMI = body mass index, ASA = American Society of Anesthesiologists, ILND = inguinal lymph node dissection, LN = lymph node, LOS = length of stay, pAJCC = pathologic staging according to the American Joint Committee on Cancer.

The utilization of partial penectomy was comparable in the two groups (80% vs. 79%; *p* = 0.88). Although indicated, ILND was omitted in 8 cases (5%), while in 16 a modified template was properly used. In 20 cases overall (10%) a minimally invasive approach was used. Overall, 57 patients (37%) experienced at least one postoperative complication but those CD grade ≥ 3 were observed in 42 (27%). Lymphangitis were the most reported adverse events (*n* = 38; 25%), followed by wound dehiscence (*n* = 34; 22%) and symptomatic lymphoceles (*n* = 33; 21%): all these were significantly more common among patients that did not achieve the Trifecta (all *p* < 0.004).

In the Kaplan–Maier analysis, the *Tetrafecta* cohort displayed significantly higher OS probabilities (Log Rank = 0.01) (Figure 2). Uni- and multivariable logistic regression analyses identified age as the only independent predictor of *Tetrafecta* achievement (OR: 0.97; 95%CI: 0.94–0.99; *p* = 0.04) (Table 2).

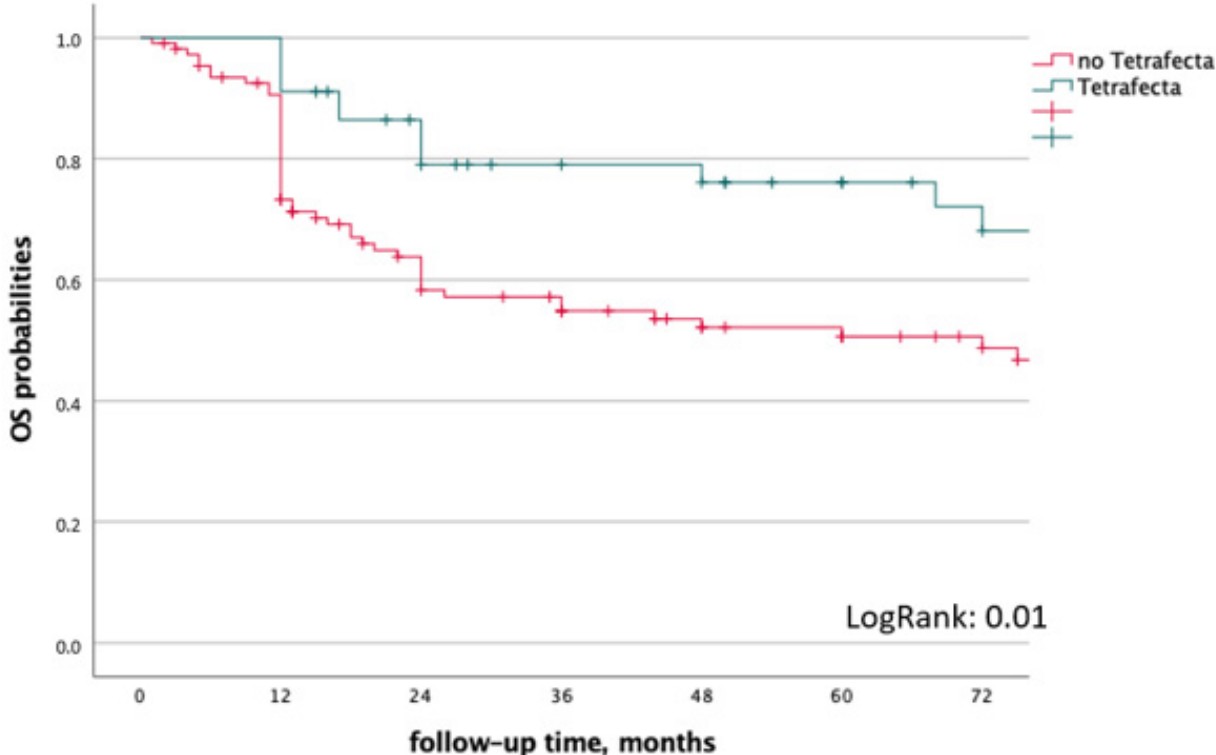

| Overall survival | 2-yr | 5-yrs |
|---|---|---|
| No Trifecta<br>- probabilities ± SE<br>- Number at risk (events) | 58±5<br>58 (38) | 51±5<br>34 (48) |
| Trifecta<br>- probabilities ± SE<br>- Number at risk (events) | 79±6<br>35 (8) | 76±7<br>22 (10) |

**Figure 2.** Kaplan–Maier analysis to assess the predictive role of Trifecta achievement on OS probabilities.

**Table 2.** Univariable and multivariable logistic regression analyses to identify predictors of *Tetrafecta* achievement.

| | | Univariable Analysis | | | | Multivariable Analysis | | |
| | | 95% CI | | | | 95% CI | | |
| | OR | Lower | Higher | *p* Value | OR | Lower | Higher | *p* Value |
|---|---|---|---|---|---|---|---|---|
| Age | 0.96 | 0.93 | 0.99 | 0.01 | 0.97 | 0.94 | 0.99 | 0.04 |
| BMI | 0.90 | 0.81 | 1.00 | 0.04 | 0.93 | 0.84 | 1.04 | 0.22 |
| ASA $\geq$ 3 | 0.998 | 0.946 | - | 0.000 | - | - | - | - |
| cN stage | | | | 0.02 | | | | 0.64 |
| - CN0 | ref. | - | - | | ref. | - | - | |
| - cN1–2 | 0.50 | 0.29 | 0.85 | 0.01 | 1.15 | 0.54 | 2.45 | 0.72 |
| - cN3 | 0.25 | 0.08 | 0.75 | 0.01 | 0.63 | 0.18 | 2.19 | 0.47 |

BMI = body mass index, ASA = American Society of Anesthesiologist.

## 4. Discussion

PC is an aggressive and mutilating disease which usually affects older adults. It is rare in western countries, accounting for <0.1% of all malignances in men [1], whilst its prevalence is higher in India, South America, and Africa [4] where the rates of neonatal circumcision are low [2]. Regardless surgical treatment, approximately one quarter of the treated men will experience disease recurrence within one year [17].

To achieve adequate oncological control at the time of PC-surgery, a 2-cm-long safe surgical margin was historically deemed mandatory [3]: this led to an over-utilization of radical penectomies and patients that received a partial amputation often experienced disfiguring cosmetic results. Recent evidence demonstrated that a 5-mm-long margin is enough to provide appropriate cancer control, so that we have witnessed a shift in the treatment paradigm toward partial penectomies. In fact, current international guidelines recommend a 3–5 mm safe margin length during partial penectomy [3]. Although this treatment paradigm shift has brought an increase in the rate of local recurrence, cancer-specific and overall survival outcomes were not jeopardized [18] (1, 4, 5). Nowadays, the vast majority of PC-patients can be offered organ-sparing treatment strategies such as circumcision, laser ablation, and wide local excision due to their early cT-stage and the distal location [19]. In case of cT2/cT3 diseases (that involve the urethra and/or corpora cavernosa) or recurrent PC, more invasive procedures such as partial or radical penectomy are required. Given the low incidence of this tumor and the advancement of organ-sparing approaches, the utilization of total amputation has decreased in recent series. More in detail, partial penectomy is indicated in case of diseases involving the shaft which cannot be treated with circumcision, laser ablation of local excision. It involves resection proximal to the tumor and terminalization of urethra and corpora. This approach is associated with a <10% recurrence rate and allows for good preservation of organ function, thus being considered the standard of care for patients with an organ-confined disease, limited to the shaft [20]. Total amputation is defined as the excision of the penis up to the suspensory ligament, preserving the proximal segments of corpora cavernosa [21], while radical penectomy entails the excision of the whole penis, with the complete removal of the two corporeal bodies. Once resection/excision phase is accomplished, a perineal urethrostomy is performed to allow micturition [21]. Penectomy is recommended in most cT3 and all cT4 staged PCs, although it may also be required in cT2 diseases if a functional residual stump is not attainable. Still, this approach could also be considered in case of cT3 tumors, if a 5 mm clear margin is achievable, given the low recurrence rates associated with this procedure and the possibility to perform radical salvage surgery [22]. Lont et al. demonstrated that, although organ-preserving approaches provide the best functional and cosmetic outcomes, partial and total penectomies are associated with a lower risk of disease recurrence (12% vs. 37%) [23]. These findings were further confirmed by a larger retrospective study based on 415 PC-patients: a 5.3% recurrence rate was observed in the partial/total penectomy group vs. 27.7% in the penile-preserving cohort, whilst survival

rates were comparable (92%) [18]. The latter observation, though, is likely confounded by the possibility to perform a more radical salvage surgery in case of tumor recurrence after circumcision, laser ablation, or local excision.

As said, although it was historically cured with radical excision, in recent years a less aggressive approach has been recommended in lower stage diseases [3], because of the disfiguring nature and psychological distress caused by the conventional treatment; however, a trend toward increased positive surgical margins (PSM) rate was observed [23]. In the United States, most pT1 (53.6%) and pT2 tumors (46.4%) undergo partial penectomy (PP), with a 7.2% PSM rate in this specific population; conversely, most pT ≧ 3 neoplasms are managed though radical amputation [24]. The former surgical strategy was the most used also in our series (*n* = 122; 79%), both for cT1 (*n* = 32) and cT2 (*n* = 90) tumors, and the PSM rate was 10% (*n* = 12/122) in this subgroup. Interestingly, even though data from the American National Cancer Database (ANCD) highlighted that positive margins are not associated with worst oncologic outcomes [19], there is evidence that patients undergoing any kind of penile-sparing therapy are at increased risk of local recurrence (37% vs. 12% at 5 years follow-up) and will probably face regional recurrence too (33% vs. 6%) [23]. More than the margin status, primary tumor size seems to significantly affect survival outcomes, being neoplasms larger than 3 cm associated with poorer OS and CSS [25,26].

Survival rates have not significantly improved since the 1990s [27] as patients often present with locally advanced diseases, failing to seek help because of the stigmata associated with PC [27]. Moreover, despite the well-known impact of ILND on survival outcomes [8,28], lymphadenectomy remains underutilized [29–32] probably because of the complication rate associated with the procedure, which ranges between 10% and 78% [33]. In fact, according to a review of the ANCD, ILND was more commonly performed in younger men, with a recent diagnosis, treated in academic centers; patients' socio-demographic factors had no effect on lymphadenectomy utilization, suggesting that the decision to perform that surgical procedure was mostly based on physician preferences [28]. Noteworthy, in the present series, ILND was omitted only in 5% of the cases while two larger retrospective analysis from the ANCD [30] and SEER (Surveillance, Epidemiology, and End Results) [34] databases pointed out an alarming 19.6–26.5% adherence to the guidelines. Conversely, according to a recent report from eight tertiary centers in Italy, the ILND utilization rate was 70% [31]. Therefore, our findings can be explained by the fact that the institutions participating to the present study are referral centers for PC and that we only included in the analysis patients with a relatively recent diagnosis.

The presence and the extent of inguinal metastases are the most important prognostic factors in PC patients. Regional lymphatic spread is always associated with a worse prognosis, being pelvic nodal involvement more threatening than inguinal node dissemination. Men with two or more inguinal metastases show a 12-fold increased risk of pelvic nodal involvement. Extracapsular growth, bilateral inguinal spread, and pelvic node disease are independent predictors of a worse disease-specific survival [35]. At 3 years, cancer-specific survival in pN0/pN1 patients is almost 100%, while it drops to 73% in case of pN2 disease [35]. There is grounded evidence that the risk of lymphatic spread is affected by PC grade, being 0–29% in well differentiated tumors vs. 33–50% in poorly differentiated ones [36]. Also cT stage is an independent risk factor for nodal metastases, with a 50–70% of N+ diseases in patients harboring a pT2 cancer vs. 50–100% in case of pT3/pT4 [35]. Extremely variable rates of nodal involvement have been reported in men with pT1 PCs, depending on the staging approach used. Nodal metastases from PC can be clinically diagnosed only in men with swollen lymph nodes in the groin: minimal lymphatic spread cannot be ruled out with physical examination, but a quarter of patients with non-palpable nodes harbors micro-metastases [35]. Also imaging studies are of limited value in the preoperative assessment of the N-stage: both computed tomography and magnetic resonance imaging are unable to identify microscopic sites of metastases, although bulky LNs may show typical radiologic signs of cancer involvement [35]. Ultrasound-guided fine-needle aspiration cytology has shown a sensitivity of 93% and specificity of 91% for palpable

lymph nodes, but only 9 of 23 metastases (sensitivity: 39%; specificity: 100%) were detected in cN0 diseases [37,38]. Periodic physical examination with exploration of the groins is still an option to manage low-risk organ-confined PCs. On the contrary, an early appropriate surgical staging and management of regional nodes is of vital importance in locally advanced cases. In fact, Lont et al. reported a 91% 3-yr disease-specific survival in patients with pT2/pT3 disease managed with dynamic sentinel node staging compared with 79% in a historical series managed by surveillance [39]. Similarly, in the largest retrospective series reported so far, Leijte et al. reported a significantly higher risk of recurrence in patients undergoing surveillance management [18].

It is assumed that PC, like other SCCs, has a tendency for locoregional growth and that a stepwise lymphogenic spread always occurs before hematogenic dissemination. It was observed that lymphatic drainage in these patients is to both inguinal sides in up to 81% of cases, but metastatic cells seem to only migrate according to a consistent pathway: skip lesions and crossover lymphatic metastases were never described [35]. Inguinal LNs represent the primary draining station. Thereafter, metastatic spread usually proceeds to the pelvic and then retroperitoneal nodes [40]. Inguinal LNs are located within the femoral (Scarpa's) triangle, which is a wedge-shaped area of the superomedial aspect of the anterior thigh, defined by the inguinal ligament (superiorly), the abductor longus muscle (medially), and the sartorius muscle (laterally). These are topographically divided into two packets: the superficial and deep nodes. The formers are located beneath the Camper's fascia and above the fascia lata, which covers the muscles of the thigh. The latter lies underneath the fascia lata, medial to the femoral vein. The two packets intercommunicate with each other and then drain into the nodes of the pelvis. Rouvière first proposed dividing the inguinal LNs into four quadrants, by drawing a cross with its middle point in the saphenous hiatus [4]. Lately, Daseler identified a fifth zone, located right at the saphenous-femoral junction [41]. A recent lymphoscintigraphic study by Leijte et al. highlighted that most of the first draining nodes are located in the superomedial segment, although individual variation exists [42]. The radical ILND consists in the resection of all lymphatic tissue located within the Scarpa's triangle, ligating the saphenous vein and exposing the femoral vessels. Although it is still considered the gold standard option to treat nodal metastases in this area, this surgery is associated with a remarkable complication rate (up to 62% in historical series). As a result, several modified templates [43,44] and minimally invasive approaches [45–51] were further proposed, with the aim to preserve oncological benefits while reducing morbidity.

Concerning surgical adequacy, our median LNs yield was 17 (IQR: 11–27), which is in line with other large series of open and minimally invasive ILNDs, where an average of 7–20 nodes were reported [52]. In fact, cadaveric studies described that ILNs range between 4 and 25 per groin, with an average of 8 in each extremity; deeper nodes are not always found, and their number is comprised between 0 and 5 [4]. Noteworthy, regardless a satisfactory median LNs yield, we failed to dissect a minimum of 7 LNs from each treated groin in 54 cases (35%), which was proposed as the standard cut-off value for regional lymphadenectomy [16].

Although a minimum number of retrieved LNs was included among the four criteria to define the *Tetrafecta* achievement, its role in determining patients' prognosis is still debated. On the contrary, LN density (defined as the ratio between positive and retrieved nodes) seems to better predict oncologic outcomes, being a value >20% associated with an increased risk of tumor recurrence/progression (HR: 2.12; 95%CI: 1.18–3.80; $p = 0.011$) [25,53].

Based on our results, more than one out of three patients ($n = 57/154$) experienced at least one post-operative complication, mostly ($n = 42/154$) CD grade $\geq$ 3. Infections were the most common adverse events, followed by wound dehiscence and symptomatic lymphoceles. Although historical studies on radical ILND reported a complication rate of 50–70%, results from modern series are consistent with our findings and the share of minor and major complications ranges between 10–78% and 0–37%, respectively [33]. As said above, as a result of the high morbidity associated with the rILND, modified templates were

further conceived to treat men with intermediate/high-risk PCs with no clinical suspicion of inguinal metastases, preserving oncological benefit while reducing complications [33]. The adoption of minimally invasive approaches to ILND was proven to provide lesser skin complications and lymphedema as compared to the conventional open approach [52]. However, the resort to video-endoscopic lymphadenectomy is still limited and, also in the present series, only 20 of these cases were observed.

Several composite outcomes have been proposed in the field of uro-oncology [10–12] though, to the best of our knowledge, our *Tetrafecta* is the first conceived to allow for a combined reporting of surgical and oncologic outcomes after PC surgery with ILND. In the present series, only 29% of patients achieved the Tetrafecta. Such a discouraging result is mainly due to the high severe complication rate (27%) and 1 year recurrence rate (24%) which, however, were consistent with the international literature [17,33]. Also our incidence of PSMs (8%) was in line with reports from the ANCD [24]. Interestingly, although adherence to guidelines was extraordinarily high and the vast majority (95%) of the included patients underwent ILND, notwithstanding a satisfactory median LNs yield ($n = 17$; IQR: 11–27) [52], approximately one out of three PC-men did not receive an adequate lymphadenectomy in both groins. Patients achieving the *Tetrafecta* displayed superior oncologic outcomes with a 5 years OS rate of $76 \pm 7\%$ compared to $51 \pm 5\%$ in non-Trifecta group (Log Rank = 0.01). At univariable logistic regression analysis, pN+ stage and older age were significantly associated with reduced probabilities of achieving the *Trifecta* (Table 2) but only the latter was confirmed as an independent predictor at multivariable analysis (OR: 0.97; 95%CI: 0.94–0.99; $p = 0.04$).

This study suffers from limitations inherent to its limited sample size and its retrospective design. First of all, despite the eighth edition of the AJCC Staging Manual now being available, TNM staging could not be retrospectively updated for our study population. Moreover, relevant histologic and genetic tumor features (such as lympho-vascular invasion and p53 overexpression) were not recorded in most patients' files and their impact on Trifecta achievement probabilities could not be assessed. Similarly, also data on systemic neoadjuvant/adjuvant treatments were missing in most records and were not included in our logistic regression model to identify predictors of our composite outcome. Furthermore, the observed Trifecta rate might not be the same if calculated on different larger series: in fact, all the institutions taking part to the present study are referral centers with a high load of locally-advanced and metastatic cases and access to experimental therapies not allowed outside clinical trials. Finally, our novel composite outcome does not take into account functional and cosmetic results of PC-surgery. In fact, these are difficult to accurately assess and, up to today, fewer evidence is available concerning the effect of penile surgery on patients' own perception of masculinity and quality of life overall [54,55]. It is likely that radical penectomy has the most detrimental effect on patient's sexual life and body image, while the vast majority of patients undergoing penile-preserving approaches are able to achieve and maintain an erection at 1 year [56]. Though, despite the detrimental effects of total penectomy on well-being with increased rates of depression and sexual anxiety, suicide rates among PC-patients are surprisingly the lowest out of all urological malignancies [57].

## 5. Conclusions

PC is an aggressive tumor, associated with a 24% risk of recurrence at 1-year follow-up. Even if most patients present with a locally advanced disease, partial penectomy represents the most common treatment option (79%), at least in referral centers. ILND is crucial for the management of this neoplasm, but a quarter of patients report severe complications.

We herein presented the first composite outcome, based on reproducible variables, specifically conceived to report results after PC with ILND. While external validation studies are awaited, this newly defined *Tetrafecta* provides a comprehensive summary of surgical and oncologic outcomes and represents a significant predictor of all-cause mortality.

**Author Contributions:** Conceptualization, A.B. (Aldo Brassetti), U.A. and G.S.; methodology, A.B. (Aldo Brassetti); validation, A.B. (Aldo Brassetti), G.S.; formal analysis, A.B. (Aldo Brassetti); investigation, U.A.; resources, A.M.B.; data curation, A.M.B., U.A., G.C., J.C., P.G., J.M.G.S., A.M.B., F.P., M.F., R.M., L.M., G.T. (Gabriele Tuderti), G.T. (Giulia Torregiani), M.C., D.C., G.M., R.V., A.B. (Alberto Breda) and O.D.C.; writing—original draft preparation, A.B. (Aldo Brassetti); writing—review and editing, G.S.; visualization, A.B. (Aldo Brassetti); supervision, G.S.; project administration, G.S. All authors have read and agreed to the published version of the manuscript.

**Funding:** This research received no external funding.

**Institutional Review Board Statement:** The study was conducted according to the guidelines of the Declaration of Helsinki, and approved on 6 September 2021 by the local Institutional Review Board (protocol code ILNDinPC—M. BBIRE. 01_060921).

**Informed Consent Statement:** Informed consent was obtained from all subjects involved in the study. Written informed consent has been obtained from the patients to publish this paper.

**Data Availability Statement:** The data presented in this study are available on request from the corresponding author.

**Conflicts of Interest:** The authors declare no conflict of interest.

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
