# Peer review of "Combined Reporting of Surgical Quality and Cancer Control after Surgical Treatment for Penile Tumors with Inguinal Lymph Node Dissection: The Tetrafecta Achievement"

_curroncol, doi:10.3390/curroncol30020146_

Round 1

Reviewer 1 Report

Dear Authors,

article was interesting, with clear methodology, valid results and its limitations clearly stated (especially retrospective design).  I would only require Authors to provide the IRB review code and appraisal date if possible

Author Response

Reviewer #1
Dear Authors, article was interesting, with clear methodology, valid results and its limitations clearly stated (especially retrospective design).
I would only require Authors to provide the IRB review code and appraisal date if possible.

We thank the reviewer for the comments.
Concerning his/her query, the present study was approved by the local ethic committee on September 6th 2021 (code: ILNDinPC – M. BBIRE. 01_060921). This detail was clarified in the dedicated “Patents” section, at the end of the manuscript:

• (Patents, page 10, lines 390-392):
The study was conducted according to the guidelines of the Declaration of Helsinki, and approved on September 6th 2021 by the local Institutional Review Board (protocol code ILNDinPC – M. BBIRE.01_060921).

Reviewer 2 Report

This study was reported the utility of the Tetrafecta achievement in patients with penile cancer who underwent surgery. Generally, this paper is well written. The reviewer thinks that this paper has useful information for readers. However, the reviewer would like to suggest some critiques to make this paper as follows.

Major revision

1.     On line 79, the subtitle should be changed from “Study objectives” to “The definition of the Ttrafecta.” This subtitle may mislead readers.

2.     Since Table 1 describes clinical T and N, the prognostic impact of these should be reviewed in the Discussion section. In the first paragraph of the Discussion section, the influence of clinical T or tumor size on prognosis should be discussed. In addition, the second paragraph should discuss the relationship between clinical N and the number and extent of lymph nodes and the number of lymph nodes to be harvested. Please cite PMID: 36005170 in your discussion.

Author Response

Reviewer #2
This study was reported the utility of the Tetrafecta achievement in patients with penile cancer who underwent surgery. Generally, this paper is well written. The reviewer thinks that this paper has useful information for readers. However, the reviewer would like to suggest some critiques to make this paper as follows.
Major revision:

1. On line 79, the subtitle should be changed from “Study objectives” to “The definition of the Ttrafecta.” This subtitle may mislead readers.

We thank the reviewer for the suggestion.
We totally agree with him/her that, in this section, we mainly defined our novel composite outcome. However, in the last line, also the aim of our paper is clarified (= assessing the ability of Tetrafecta to predict OS probabilities).
As suggested, the subtitle of this section and related manuscript were modified.

• (Materials and Methods, page 2, lines 86-91)

2.2 Tetrafecta definition and study objective
The Tetrafecta for PC-surgery was conceived combining four standardized and reproducible variables: negative local surgical margins (NSM), no complications CD grade ≥3, ≥ 7 LNs retrieved from each treated groin[16], no evidence of disease at 12 months (NED12mo). We assessed the ability of this novel composite outcome to predict OS probabilities.

2. Since Table 1 describes clinical T and N, the prognostic impact of these should be reviewed in the Discussion section. In the first paragraph of the Discussion section, the influence of clinical T or tumor size on prognosis should be discussed. In addition, the second paragraph should discuss the relationship between clinical N and the number and extent of lymph nodes and the number of lymph nodes to be harvested. Please cite PMID: 36005170 in your discussion.

We thank the reviewer for the suggestions. The recommended paper [Clinical Lymph Node Involvement as a Predictor for Cancer-Specific Survival in Patients with Penile Squamous Cell Cancer] was carefully red and used to ameliorate the Discussion of our manuscript; it was obviously cited too.

• (Discussion, page 6, lines 186-219)
Although it was historically treated with radical excision, in recent years a less aggressive approach has been recommended in lower stage diseases [2] but a trend towards increased positive surgical margins (PSM) rate was observed[18]. In the United States, most pT1 (53.6%) and pT2 tumors (46.4%) undergo partial penectomy (PP), with a 7.2% PSM rate in this specific population; conversely, most pT≧3 neoplasms are managed though radical amputation[19]. This surgical strategy was the most used also in our series (n= 122; 79%), both for cT1 (n=32) and cT2 (n=90) tumors, and the PSM rate was 10% (n=12/122) in this subgroup. Interestingly, even though data from the American National Cancer Database (ANCD) highlighted that positive margins are not associated with worst oncologic outcomes[19], there is evidence that patients undergoing any kind of penile-sparing therapy are at increased risk of local recurrence (37% vs 12% at 5 years follow-up) and these latter will more probably face also regional recurrence (33% vs 6%)[18]. More than the margin status, primary tumor size seems to significantly affect survival outcomes, 
being neoplasms larger than 3 cm associated with poorer OS and CSS [10.1007/s12672-021-00416-7][10.3390/curroncol29080432]

Round 2

Reviewer 2 Report

none.